# Compared to Individuals with Mild to Moderate Obstructive Sleep Apnea (OSA), Individuals with Severe OSA Had Higher BMI and Respiratory-Disturbance Scores

**DOI:** 10.3390/life11050368

**Published:** 2021-04-21

**Authors:** Leeba Rezaie, Soroush Maazinezhad, Donald J. Fogelberg, Habibolah Khazaie, Dena Sadeghi-Bahmani, Serge Brand

**Affiliations:** 1Sleep Disorders Research Center, Kermanshah University of Medical Sciences, Kermanshah 67146, Iran; rezaie.phd.ot@gmail.com (L.R.); smedical90@gmail.com (S.M.); dena.sadeghibahmani@upk.ch (D.S.-B.); 2School of Nursing and Midwifery, Kermanshah University of Medical Sciences, Kermanshah 67146, Iran; 3Department of Rehabilitation Medicine, University of Washington School of Medicine, Seattle, WA 98195, USA; fogelber@uw.edu; 4Departments of Physical Therapy, University of Alabama at Birmingham, Birmingham, AL 35209, USA; 5Center for Affective, Stress and Sleep Disorders (ZASS), Psychiatric University Hospital Basel, 4002 Basel, Switzerland; 6Department of Clinical Research, University of Basel, 4031 Basel, Switzerland; 7Substance Abuse Prevention Research Center, Health Institute, Kermanshah University of Medical Sciences, Kermanshah 67146, Iran; 8School of Medicine, Tehran University of Medical Sciences, Tehran 25529, Iran; 9Division of Sport Science and Psychosocial Health, Department of Sport, Exercise and Health, University of Basel, 4052 Basel, Switzerland

**Keywords:** anthropometry, obstructive sleep apnea, polysomnography, severity

## Abstract

Objective: Individuals with obstructive sleep apnea (OSA) are at increased risk to suffer from further somatic and sleep-related complaints. To assess OSA, demographic, anthropometric, and subjective/objective sleep parameters are taken into consideration, but often separately. Here, we entered demographic, anthropometric, subjective, and objective sleep- and breathing-related dimensions in one model. Methods: We reviewed the demographic, anthropometric, subjective and objective sleep- and breathing-related data, and polysomnographic records of 251 individuals with diagnosed OSA. OSA was considered as a continuous and as categorical variable (mild, moderate, and severe OSA). A series of correlational computations, X^2^-tests, F-tests, and a multiple regression model were performed to investigate which demographic, anthropometric, and subjective and objective sleep dimensions were associated with and predicted dimensions of OSA. Results: Higher apnea/hypopnea index (AHI) scores were associated with higher BMI, higher daytime sleepiness, a higher respiratory disturbance index, and higher snoring. Compared to individuals with mild to moderate OSA, individuals with severe OSA had a higher BMI, a higher respiratory disturbance index (RDI) and a higher snoring index, while subjective sleep quality and daytime sleepiness did not differ. Results from the multiple regression analysis showed that an objectively shorter sleep duration, more N2 sleep, and a higher RDI predicted AHI scores. Conclusion: The pattern of results suggests that blending demographic, anthropometric, and subjective/objective sleep- and breathing-related data enabled more effective discrimination of individuals at higher risk for OSA. The results are of practical and clinical importance: demographic, anthropometric, and breathing-related issues derived from self-rating scales provide a quick and reliable identification of individuals at risk of OSA; objective assessments provide further certainty and reliability.

## 1. Introduction

Obstructive sleep apnea (OSA) is one of the most common breathing-related sleep disorders. The American Academy of Sleep Medicine [1] defines OSA as the repetitive and complete or partial cessation of airflow for 10 or more seconds during sleep. Repetitive and complete or partial upper-airway obstructions lead to such cessation of airflow. Further, the polysomnographic record (PSG) should show five or more predominantly obstructive respiratory events per hour (obstructive and mixed apneas, hypopneas or respiratory effort-related arousals (RERAs)). Relatedly, OSA is diagnosed if PSG analyses record 15 or more predominantly obstructive respiratory events (apneas, hypopneas, or RERAs) per hour [1].

Regarding prevalence rates of OSA, a meta-analysis reported the following figures: 3% of women and 10% of men aged 30 to 49 years have moderate to severe OSA; and 9% of women and 17% of men aged 50 to 70 years have moderate to severe OSA [2]. Population-based studies reported prevalence rates of 17% to 23.4% for women and 22% to 49.7% for men [3,4]. Differences in the clinical and subjective diagnoses may explain the large range of prevalence rates. At a behavioral level, OSA is associated with habitual snoring, nonrestorative sleep (most probably due to frequent arousals and sleep fragmentation), headache, daytime sleepiness, excessive fatigue, cognitive impairment, and increased frequency of car accidents [5,6,7]. At a physiological level, OSA and OSAS (Obstructive Sleep Apnea Syndrome) are associated with intermittent hypoxemia, higher serum and plasma interleukin-6 levels [8], and higher serum and plasma TNF-alpha levels [9], always compared to healthy controls. Similarly, OSA appeared to be associated with autoimmune rheumatic disease [10]. Next, OSA appeared to be an independent factor for cardiovascular diseases [11,12,13,14] and all-cause mortality [14], while further associations were observed among the occurrence of OSA and diabetes mellitus [15,16,17,18,19] and hormonal disorders [20]. For morning saliva and blood cortisol levels, compared to adult healthy controls, adults with OSAS had no difference in cortisol levels [21]. In contrast, compared to healthy controls, children with OSAS had statistically significant lower cortisol levels [21]. Although salivary morning cortisol levels in adults with OSAS did not differ significantly from those of healthy controls, children with OSAS had significantly lower cortisol concentrations than control subjects. These findings are clinically significant, given the association often seen between diminished cortisol levels and chronic psychophysiological stress [22,23]. Overall, lowered cortisol levels and higher inflammatory markers appear to [20] mirror an increased psychophysiologically stressed organism with OSA, when compared to healthy controls. Further, previous reports revealed that OSA was associated with the risk of mood disorder, cognitive disorder, systemic hypertension, coronary artery disease, stroke, congestive heart failure, or atrial fibrillation [24,25,26,27]. Clearly, the diagnosis and treatment of OSA should be considered as a health priority.

The latter observation holds particularly true, as individuals with OSA have a substantial economic impact on the healthcare system [28]; individuals with severe OSA, understood as the costliest and sickest upper third among all individuals with OSA, caused 65–82% of all medical costs, mainly due to cardiovascular diseases tightly related to OSA.

Next, epidemiological studies have shown that demographic and anthropometric dimensions such as male gender, higher age, and higher body weight increase the risk of OSA [27,29].

The gold standard to diagnose OSA is the polysomnographic assessment (PSG); the main metric is the apnea/hypopnea index (AHI), which reflects the average number of breathing disturbances per hour [1,19]. Accordingly, an AHI > 5 is considered as OSA and classified in mild, moderate, and severe OSA based on the increasing AHI index [1,30].

Treatment decisions for OSA are largely based on OSA severity. Continuous positive airway pressure (CPAP) devices are indicated in individuals with concomitant symptoms and AHI ≥ 5, and individuals with no symptoms with AHI ≥ 15. CPAP devices reduce OSA severity and excessive daytime sleepiness, and improve quality of life [31,32].

In two representative studies of PSG-derived sleep continuity and sleep architecture indices in individuals with OSA [33,34], the following patterns were observed: for sleep continuity, higher sleep fragmentation was reported [33,34], along with a faster decay of deep sleep, compared to healthy controls [34]. Sleep efficiency decreased in an almost linear fashion from healthy controls to individuals with mild, moderate, and severe OSA [33], with individuals with severe OSA having the lowest sleep efficiency, compared to individuals with no, mild, and moderate OSA, and always assessed with sleep-EEG. In parallel, arousal index and wake index were highest in individuals with moderate and severe OSA, compared to individuals with no and mild OSA [33].

Regarding the research on OSA in Iran, data are less abundant. Prevalence rates of OSA in Iranian adults range from 5% to 38.9% [35]. Further, to identify individuals at high risk of OSA, data were mainly based on self-rating questionnaires such as the Berlin Questionnaire [5,36] or the STOP-Bang [35]. Foroughi et al. [37] reported in their brief review the following predictors: snoring, male gender, age over 50 years, hypertension, and metabolic diseases. To our knowledge, except the recently published data on sleep spindles among individuals with OSAS [33], PSG data in combination with demographic and anthropometric information of Iranian individuals with OSA are missing. To counter this, in the present study, we blended demographic, anthropometric, and PSG data to predict OSA severity. Accordingly, the key questions of the present study were whether and which demographic, anthropometric, and PSG-related dimensions could predict OSA severity. Given the explorative character of the model, no hypotheses were formulated.

## 2. Material and Methods

### 2.1. Study Design

In this retrospective observational study, medical records of 650 individuals who underwent overnight PSG at the Sleep Disorders Research Center of Kermanshah University of Medical Sciences (Kermanshah Iran) in 2012–2019 were reanalyzed. Inclusion criteria for data analysis were: (1) age between 20 and 80 years; (2) complete PSG data; (3) complete medical records; and (4) signed written informed consent to use data for research purposes. All records and data were fully anonymized. Exclusion criteria were: sleep disorders that could be explained as the result of psychiatric or somatic issues, and as ascertained by an experienced psychiatrist and expert in sleep research. The ethical committee of Kermanshah University of Medical Sciences (Kermanshah, Iran; code: IR.KUMS.REC.1398.1032) approved the study, which was performed in accordance with the seventh and current version [38] of the Declaration of Helsinki.

Of the 650 data sets screened, 399 (61.4%) were excluded, and 251 (38.6%) were analyzed (see Figure 1).

### 2.2. Measures

#### 2.2.1. Demographic and Anthropometric Data

The demographic data were: age (years); gender (male, female); tobacco smoking (yes vs. no). The objectively measured anthropometric data were: height (cm); weight (kg); neck circumference (cm); waist circumference (cm).

#### 2.2.2. Sleep Quality

Participants completed the Persian version of the Pittsburgh Sleep Quality Index [39]. The Persian version was psychometrically validated [40,41,42,43]. The PSQI is a self-report scale that is completed in 5 min. It consists of 19 items and contains seven subscales (subjective sleep quality, sleep latency, sleep duration, sleep efficiency, sleep disturbance, sleeping medication, daytime dysfunction), each weighted equally on a scale from 0 to 3, with higher scores indicating poorer sleep quality. The seven components are then summed to obtain an overall PSQI score, ranging from 0 (good sleep quality) to 21 (poor sleep quality). Total scores of ≥ 5 reflect poor sleep, associated with considerable sleep complaints (Cronbach’s α: 0.85).

#### 2.2.3. Daytime Sleepiness

As described in another study [44], to assess daytime sleepiness, participants completed the Farsi version [45] of the Epworth Sleepiness Scale (ESS) [46]. The ESS consists of eight items rating the odds of dozing off during different activities. Answers are given on 4-point rating scales (0–3), with higher-sum scores reflecting greater daytime sleepiness. The sum score ranges from 0 to 30 points; a global score greater than 10 indicates excessive daytime sleepiness. ESS scores were treated as continuous (Cronbach’s α: 0.86).

#### 2.2.4. Self-Reported Breathing Issues; STOP-Bang Questionnaire

As described in another study [47], participants completed the Farsi version of the STOP-Bang questionnaire [48,49], a self-report, forced-choice (yes/no) scale to rate the risks of suffering from OSA. The “STOP” portion of the questionnaire consists of four questions related to snoring (S), tiredness during the daytime (T), observed apneas (O), and high blood pressure (P). When two or more questions are answered with yes, then the person has a higher risk for obstructive sleep apnea/hypopnea syndrome (OSAHS). The “Bang” portion is evaluated by assessing BMI > 35 kg/m^2^ (B), age (>50 years) (A), neck circumference (>40 cm) (N), and gender (male) (G). One point is assigned for each positive answer, and zero for each negative answer. High risk for the OSAHS on the STOP-Bang is indicated when three or more of the eight questions are answered yes. The total score was treated as a continuous variable. The STOP-Bang is a sensitive, reliable screening tool for OSA, frequently used in outpatient sleep clinics.

#### 2.2.5. Polysomnography and Breathing-Related Information 

All participants underwent an overnight PSG. The assessments were as follows: Three electroencephalogram (EEG) channels (frontal, central, and occipital); electro-oculogram (EOG); electromyogram (EMG); airflow (by oronasal thermistor and nasal air pressure transducer); thoracic and abdominal respiratory effort (induction plethysmography); peripheral oxygen saturation (SpO_2_); and body position. PSG data were analyzed following the guidelines of the American Academy of Sleep Medicine [1]. The parameters were: sleep continuity—total sleep time (TST; h), wakenings after sleep onset (nr), and sleep efficiency (%); sleep architecture—stage N1 (%), stage N2 (%), stage N3 (%), and REM stage (%); breathing-related dimensions—AHI (events/hour), RDI (respiratory disturbance index; events/hour), mean SpO_2_ (%), and snoring index. Further, the AHI index was calculated and used for diagnosis of OSA and OSA severity; the following cut-off values were employed: non-OSA (AHI < 5/h), mild OSA (5 ≤ AHI ≤ 15), moderate OSA (15 ≤ AHI ≤ 30), and severe OSA (AHI ≥ 30) [30].

#### 2.2.6. Further Medical Complaints

The following data were taken from the medical records: tobacco-smoking status (yes vs. no); hypertension (yes vs. no); diabetes (yes vs. no); cardiovascular diseases (yes vs. no).

#### 2.2.7. Statistical Analysis

Single Kolmogorov–Smirnov tests showed that continuous variables were normally distributed.

A series of Pearson’s correlations was performed to calculate the associations between age, anthropometric information, subjective and objective sleep dimensions, and objective breathing-related data.

A series of X^2^-tests was performed to calculate the distribution of AHI status (low, moderate, severe) between gender, tobacco-smoking status (yes. vs. no), hypertension (yes. vs. no), diabetes (yes. vs. no), and cardiovascular diseases (yes. vs. no).

Using a series of ANOVAs, we tested whether age, anthropometric information, subjective and objective sleep dimensions, and objective breathing-related data differed between participants with mild, moderate, and severe OSA.

Effect sizes for F-tests were reported as partial eta-squared [$\eta_{p}^{2}$]; the cut-off values were: $\eta_{p}^{2}$ < .019 = trivial effect size (T); $.02<\eta_{p}^{2}$ < .059 = small effect size (S); .06 < $\eta_{p}^{2}$ < .139 = medium effect size (M); $\eta_{p}^{2}$ > .14 = large effect size (L).

To predict AHI (continuous variable), a multiple regression analysis was performed with AHI as dependent variable, and age, anthropometric dimensions, and subjective and objective sleep dimensions as predictors. Preliminary conditions to perform a multiple regression analysis were met [50,51,52]: N = 251 > 100; predictors explained the dependent variable (R = .971, R2 = .943; the number of predictors was (18) × 10 = 180 < N (251), and the Durbin–Watson coefficient was between 1.5 and 2.5, indicating that the residuals of the predictors were independent.

An alpha level of < .05 was considered as statistically significant.

All statistical calculations were performed with SPSS^®^ version 25.0 (IBM Corporation, Armonk, NY, USA) for Apple Mac^®^.

## 3. Results

### 3.1. General Observations

Table 1 provides the descriptive overview of participants’ demographic, anthropometric, and health-related information.

Briefly, 178 (70.9%) out of 251 participants were males; participants’ mean age was 49.50 years; mean BMI was 30.18; and concomitant medical complaints were cardiovascular diseases, diabetes, and hypertension. The majority did not smoke tobacco. Typically, participants complained about snoring, insomnia, and excessive daytime sleepiness.

### 3.2. Associations between Age, BMI, Subjective and Objective Sleep Parameters, and Subjective and Objective Breathing-Related Parameters

Table 2 (split in Table 2A,B) provides the descriptive statistics and correlation coefficients between age, BMI, subjective and objective sleep parameters, and subjective and objective breathing-related parameters.

A higher age was associated with a higher BMI, a higher risk of OSAS (self-rated), a shorter sleep duration, more awakenings after sleep onset, a lower sleep efficiency, higher AHIs and RDIs, and a lower peripheral oxygen saturation.

A higher BMI was associated with higher neck and waist circumferences, a higher daytime sleepiness, a lower self-rated sleep quality, a higher risk of OSAS (self-rated), a lower sleep efficiency, a lower N3 sleep (in %), more REM sleep (in %), higher AHIs and RDIs, a lower peripheral oxygen saturation, and a higher snoring index.

A higher neck circumference was associated with a higher waist circumference, a higher daytime sleepiness, a lower self-rated sleep quality, a longer N1, a shorter N2 and REM sleep (always in %), and higher AHIs and RDIs.

A higher waist circumference was associated with a higher risk of OSAS (self-rated), a shorter N3 (in %), higher AHIs and RDIs, a lower peripheral oxygen saturation, and a higher snoring index.

A higher daytime sleepiness was associated with a higher risk of OSAS (self-rated), higher AHIs and RDIs, and a lower peripheral oxygen saturation.

A self-rated lower sleep quality was associated with more REM sleep (in %).

A higher risk of OSAS (self-rated) was associated with higher AHIs and RDIs, a lower peripheral oxygen saturation, and a higher snoring index.

A shorter sleep duration was associated with more awakenings after sleep onset, a lower sleep efficiency, more N1 and N3, less N2 and REM sleep (always in %), a lower peripheral oxygen saturation, and a higher snoring index.

A higher number of awakenings after sleep onset was associated with a lower sleep efficiency, less N3 and more REM sleep, a lower peripheral oxygen saturation, and a higher snoring index.

A higher sleep efficiency was associated with more N1 and N3 sleep, lower N2 and REM-sleep (always in %), lower AHIs, and a higher peripheral oxygen saturation. More N1 was associated with less N2 and REM sleep (always in %). More N2 was associated with less N3 (in %). More N3 was associated with less REM sleep (in %) and lower AHIs and RDIs. More REM sleep was associated with higher AHIs and RDIs and a lower snoring index.

A higher AHI was associated with higher RDIs, a lower peripheral oxygen saturation, and a higher snoring index.

A higher RDI was associated with a lower peripheral oxygen saturation and a higher snoring index.

Overall, the pattern of results was such that higher anthropometric indices were associated with more unfavorable subjective and objective sleep indices and with higher objective breathing-related sleep issues.

### 3.3. Apnea/Hypopnea Index (Categorial), Gender, Tobacco-Smoking Status, Diabetes, Hypertension, and Cardiovascular Diseases (Categorical Variables)

The AHI was split into mild, moderate, and severe. Table 3 provides the descriptive statistical indices. Gender, tobacco-smoking status, and diabetes status were unrelated to AHI categories. More cases of hypertension and cardiovascular diseases were observed among individuals with severe OSA.

### 3.4. Apnea/Hypopnea Index (Categorial) and Demographic, Anthropometric, and Sleep-Related Dimensions

Table 4 provides the descriptive and inferential statistical overview of demographic (age), anthropometric (BMI, neck circumference, waist circumference) subjective (daytime sleepiness, sleep quality, risk of OSAS) and objective (sleep-EEG indices: sleep- and breathing-related (RDI, SpO_2_, snoring index)) dimensions, shown separately for individuals with mild, moderate, and severe AHI.

Compared to individuals with mild or moderate OSA, individuals with severe OSA were older and had a higher BMI, a larger neck and waist circumference, higher self-rated OSA, a higher Respiratory-Disturbance Index, more REM sleep, a lower SpO_2_, and a higher snore index.

No meaningful mean differences were found for subjective sleep disturbances (PSQI), daytime sleepiness, stage N1 and N2 (%), total sleep time, sleep efficiency, or wake index.

### 3.5. Predicting Apnea/Hypopnea Index (Continuous Variable) by Sociodemographic, Anthropometric, and Subjective and Objective Sleep- and Breathing-Related Sleep Variables

Preliminary conditions to perform a multiple regression analysis were met: N = 251 > 100; predictors explained the dependent variable (R = .971, R2 = .943); the number of predictors was (18) × 10 = 180 < N (251), and the Durbin–Watson coefficient was between 1.5 and 2.5, indicating that the residuals of the predictors were independent.

Table 5 provides the statistical overview of the multiple regression. A higher respiratory index, a longer N2 (%), and a shorter total sleep time (h) predicted a higher apnea/hypopnea index, while the following predictors were excluded from the equation, as they did not reach statistical significance (all ts < 1.0; ps > .30): age, gender, cardiovascular diseases, hypertension, BMI, neck and waist circumference, daytime sleepiness, subjective OSA (STOP-Bang), subjective sleep disturbances (PSQI), sleep efficiency, average SpO_2_, snore index, REM sleep, and N1 and N3 sleep (%).

## 4. Discussion

The key results found in the present medical chart reports were that higher obstructive sleep apnea (OSA) scores (continuous dimensions; Table 2A,B) were associated with a broad range of unfavorable dimensions of anthropometric and subjective and objective sleep- and breathing-related parameters. When OSA is considered as a categorical variable, individuals with severe OSA were older and had the highest BMI, the largest neck and waist circumference, the highest subjective OSA scores, the longest REM stage (%), the highest snore index, the lowest mean SpO_2_ (%), and, most importantly, the highest Respiratory-Disturbance Index (RDI), always compared to individuals with mild or moderate OSA (Table 4). Finally, when all dimensions were entered into the multiple regression model, a higher RDI, a longer N2 sleep (%), and a shorter total sleep time predicted higher OSA scores (Table 5). In our opinion, the present pattern of results expands upon the current literature in the following ways. First, the sample was quite large; second, a broad variety of different sources of data were considered, ranging from demographic, anthropometric, and subjective sleep- and breathing-related data to objective sleep- and breathing-related data, and in doing so, several independent predictors were taken into consideration; third, AHI was used as continuous and categorical variable, and with this, a broader variety of statistical models could be tested; fourth, to our knowledge, this is the first comprehensive study performed on this topic in the Iranian area.

Given the complexity and the heterogeneity of results reported in the current literature of this field, no straightforward hypotheses were formulated. Nevertheless, aspects of the present pattern of results were comparable with previous findings. These aspects are considered in turn.

Regarding gender, consistent with previous studies [2,3,4,27,29], more males than females were assessed in the center; the male-to-female ratio was 2.44:1.

Next, regarding age, older age was observed among individuals with severe OSA (see Table 4), which again reflected previous results [2,3,4].

Larger neck and waist circumferences were considered proxies of higher BMI indices, and higher BMI indices were observed among individuals with severe OSA; as such, the present results (see Table 4) match what we also know from other studies [27,29].

Regular physical activity is considered a key intervention to reduce body weight, raising the question of whether regular physical activity interventions could favorably impact OSA indices. Research on this question has yielded mixed results; while some studies suggest regular physical activity as an evidence-based method to reduce OSA indices via decreased body weight [53], others were more cautious [54,55,56]. However, given that regular physical activity is considered the “medicine” for a broad range of somatic [57,58,59,60,61,62,63,64,65] and psychological complaints [66,67,68,69], including insomnia and further sleep complaints [70,71,72,73,74,75,76,77,78], applying appropriately designed regimens of physical activity in OSA treatment protocols would be warranted if there are no significant contraindications.

Next, a higher incidence of cardiovascular diseases and hypertension were observed among individuals with severe OSA. Following Tarasiuk and Reuveni [28], individuals with OSA have a high economic impact on healthcare systems, and this appeared to be largely due to the concomitant comorbidity with cardiovascular issues, including hypertension. Specifically, Dredla and Castillo [26] reported in their systematic review that OSA was a risk factor for developing hypertension and cardiovascular disease, and that there is a link between untreated moderate to severe OSA and hypertension, congestive heart failure, coronary artery disease, and cardiac arrhythmia. Given this, early diagnosis and treatment of OSA is critically important in preventing further adverse consequences of OSA. As mentioned elsewhere, the first line treatment of OSA is the use of continuous positive air pressure (CPAP) devices, though at least two issues were observed. First, results from eight randomized control trials showed that the use of CPAP devices was unlikely to reduce the risk of cardiovascular issues [79]. Second, CPAP adherence is an issue [25]. In contrast, there is extant evidence that psychotherapeutic relaxation techniques and yoga have the potential to reduce hypertension [80,81,82] (but see also [83] for opposite results). Indeed, the combination of patient education and progressive muscle relaxation improved adherence to using CPAP devices in individuals with OSA.

Our results showed that smoking was reported in 23.5% of our sample, but smoking was not associated with increased AHI severity category. Previous research has established that smoking can negatively affect certain physiological sleep parameters by increasing sleep latency and reducing nocturnal oxygen saturation [84]. This result was not consistent with findings of previous studies that smoking was related to OSA severity [85,86], but was in accordance with other negative findings [87,88]. Despite the divergent findings about the relationship between OSA severity and smoking, the well-established detrimental effects of smoking, including the increased risk of cardiovascular and respiratory diseases, highlight the importance of smoking cessation.

Next, the most common complaint among study participants was snoring (75.6%). The correlation of snoring and OSA is well known, and it is a symptom frequently noted by family members of those affected. In the present study, more than of 2/3 of our patients complained of snoring, and this may be one of the main factors that led them to seek treatment from a physician. Furthermore, the results showed that the snore index (total number of snores/total sleep time) increased with OSA severity. While the quality of the present data does not provide any evidence, an association between larger uvula size, higher snoring, and higher OSA is conceivable [89].

Regarding insomnia, this was the second most common complaint among our sample (12.3%), although only 15 (5.97%) of them met diagnostic criteria for insomnia. The difference between the percentage of those complaining of insomnia and those with a formal insomnia diagnosis can be attributed to the discrepancy of subjective and objective sleep estimation [90]. Comorbidity of OSA and insomnia has been reported in previous studies. Insomnia may develop in patients with OSA due to repeated awakening during sleep, resulting in dysfunctional sleep behaviors. In addition, sleep fragmentation in patients with insomnia may worsen OSA by reducing non-rapid eye movement (non-REM) sleep [91,92]. As indicated by these studies, since both insomnia and OSA are the risk factors for cardiovascular disease, assessment of insomnia in patients with OSA versus OSA in patients without insomnia is necessary.

Our results also showed that there was no significant difference found among the three groups of OSA and sleep quality; i.e., we did not find poorer sleep quality in patients with moderate to severe OSA. This result is inconsistent with a previous study, in which higher sleep quality correlated with lower OSA severity [93]. In our study, the mild group had the worst sleep quality, suggesting that factors other than OSA are impacting sleep quality for this group.

Regarding the relationship between AHI and daytime sleepiness, results were complex. While there was a negative association between higher AHI and higher daytime sleepiness (Table 2A,B), such a pattern was not observed when AHI categories were used. Daytime sleepiness did not statistically differ between individuals with mild, moderate, and severe AHI. As such, it appears that the predictive value of daytime sleepiness, as assessed with the ESS, should be considered with caution. This claim matches well with the observation that CPAP therapy does not mitigate daytime sleepiness; given this, there is a need for a combination of different kinds of interventions to treat daytime sleepiness [94].

The results of the PSG parameters studied showed that there were significant differences with regard to AHI and RDI among the three groups, while the associations with sleep stages and REM sleep were modest (Table 2A,B; Table 4). As such, it appears that the key outcome variable of polysomnographic recordings is RDI (see also Table 5 for multiple regression analysis). Given this, to improve respiration, CPAP devices are the first-line treatment intervention [95].

Our results showed that OSA was associated with changes in sleep architecture. The three groups of OSA severity had no significant difference with regard to REM latency, although the severe OSA group had the greatest REM duration. These results can be explained by the physiological changes during REM sleep that can lead to upper airway pressure collapse and AHI events [96]. Aurora et al. reported that OSA that occurred during REM sleep may be a marker of greater underlying cardiovascular disease. This issue is important for managing patients with cardiovascular disease who have been diagnosed with OSA, since CPAP therapy may not effectively treat their underlying disease [97]. In addition, the severe OSA group had shorter N3-stage sleep and proportionally more light sleep, which was in accordance with the higher arousal index seen in severe OSA.

Our results showed that total sleep time was reduced by OSA severity (large effect size). The reductions of total sleep time in patients with OSA may be an explanation for some daytime consequences of OSA, including higher daytime sleepiness, and the negative effect on mood, cognitive function, and quality of life. This underscores the importance of early treatment of OSA.

Finally, our results showed that there was a significant difference in mean SPO_2_ among those with severe and those with mild OSA, with the severe group having lower mean SPO_2_. The reduction of SPO_2_ in patients with OSA may cause reduction in tidal volume, intermittent arterial hypoxemia, and hypercapnia. Compensatory responses of the respiratory and sympathetic nervous systems result in peripheral vasoconstriction, depressed myocardial contractility, oxidative stress, inflammation, and endothelial dysfunction. This response is associated with cardiovascular disease, hypertension, and cerebrovascular accident [98,99]. Considering the association between OSA and both cardiovascular disease and hypertension, as well as the correlated mortality and morbidity, the critical importance of early diagnosis and treatment of OSA becomes clear.

The results from the multiple regression analysis (Table 5) demand particular attention, as the pattern of predictors was somehow unexpected: A lower objectively measured sleep duration, more N2 sleep (%), and a higher Respiratory-disturbance Index predicted higher AHI scores. We note that demographic, anthropometric, and subjective sleep- and breathing-related indices were excluded from the equation, as they did not reach statistical significance. It follows that these results would imply that mere objectively assessed dimensions would be clinically useful. However, these results are critical in the following ways. First, by default, multiple regression models are used to minimizing both the number of predictors and the error variance. As such, only predictors able to contribute substantively to the variance of the outcome variable are considered. Given this, and given that stepwise procedures are highly discouraged [100], it is conceivable that the present pattern of results is biased due to issues of multicollinearity and suppression [50,52]. Overall, we suggest considering results from the multiple regression analysis with caution.

Despite the novelty of the results, several limitations should be considered. First, while we were able to blend demographic, anthropometric, and subjective and objective sleep- and breathing-related data, no neurophysiological parameters such as inflammatory markers (e.g., interleukin-6 (IL-6), tumor necrosis factor alpha (TNF-α), or cortisol levels were gathered. Neurophysiological parameters could be understood as proxies of an organism under acute and chronic stress conditions. As such, it would have been interesting to understand participants’ current and past physiological stress status. Second, the cross-sectional data set does not contain data of individuals with OSA following a treatment. The first-line treatment of OSA is the use of CPAP devices; a recent systematic review and meta-analysis [101] of four randomized clinical trials among individuals with OSA and older than 65 years showed that compared to control or sham conditions, CPAP improved dimensions of daytime sleepiness, emotion regulation, symptoms of depression and anxiety, and neurocognitive performance. In contrast, results from eight randomized control trials showed that the use of CPAP was unlikely to reduce the risk of cardiovascular issues [79]. Third, only data of individuals with OSA referred to the Sleep Disorders Research Center of the Kermanshah University of Medical Sciences (KUMS; Kermanshah, Iran) were available. This has two implications. First, a systematic sample bias cannot be ruled out; second, it is conceivable that data from individuals with undiagnosed OSA living in the general population might have yielded another pattern of results. Third, exclusively data of adults were analyzed, while children with OSA were at increased risk of a chronically stressed psychophysiological mechanism, as reflected in a continuously decreased cortisol secretion [21]. Next, a more multifaceted analysis, such as phenotyping individuals with OSA, might have yielded further results [102]. However, for the following reasons, we desisted from such analyses. First, such fine-grained analyses were not in the scope of the present study. This holds particularly true as regards more complex analyses such as craniofacial morphology and chemoreflex sensitivity, which require a sophisticated clinical validation. Second and relatedly, such validations, along with the integration of phenotyping strategies of genomic, molecular, cellular, and clinical data, might be important at an academic level and for fundamental research, while such efforts should be balanced against the practical and clinical importance and feasibility. Last, the assessment of a gender- and age-matched group of healthy controls would have allowed us to compare if and to what extent sleep characteristics of the present participants with OSA were similar or dissimilar to healthy individuals.

## 5. Conclusions

Individuals with OSA have a substantial economic impact on healthcare systems [28]. Identifying individuals with OSA is thus important to decrease their health issues and to reduce healthcare costs. The present pattern of results suggests that the thorough assessment consisting of demographic, anthropometric, and subjective and objective sleep- and breathing-related dimensions (see Table 2A,B and Table 3) yield the most reliable results. This is important because the results of the regression model suggest that exclusively data from objective sleep (i.e., polysomnographic assessment, including sleep-disordered breathing issues such as a higher respiratory disturbance index and a higher snoring index) appeared to be the best predictors. However, at a clinical and practical level (Table 2), subjective sleep-disordered breathing (STOP-Bang), along with male gender and higher BMI, also offer the opportunity for nonspecialized healthcare providers to identify individuals at increased risk of OSA. Importantly, general self-rating questionnaires such as the PSQI for sleep quality and the ESS for daytime sleepiness appeared to be too coarse-grained to distinguish OSA severity.

## Figures and Tables

**Figure 1 life-11-00368-f001:**
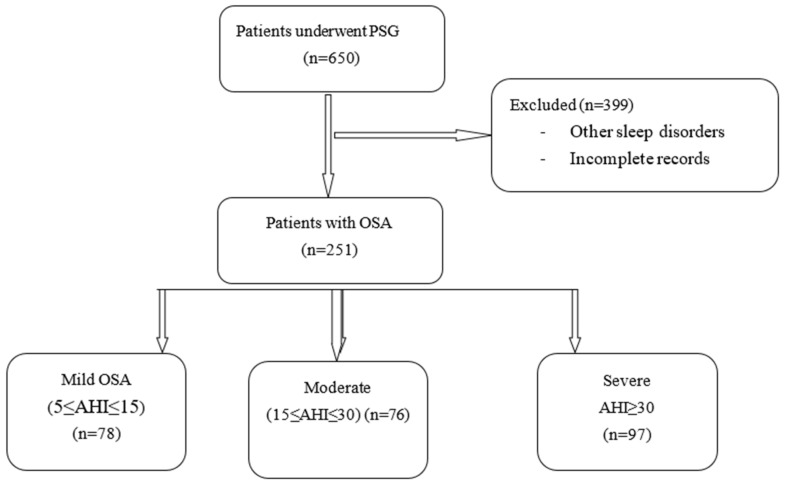
Study flowchart. PSG: polysomnography; OSA: obstructive sleep apnea.

**Table 1 life-11-00368-t001:** Demographic and anthropometric characteristics of OSA patients.

Variables	Mean (SD)
Age (years)	49.50 (12.70)
Height (cm)	170.92 (9.90)
Weight (kg)	88.18 (15.63)
BMI	30.18 (5.07)
Neck circumference (cm)	40.23 (3.58)
Waist circumference (cm)	104.17 (13.14)
**Gender**	**Frequency (%)**
Female	73 (29.1)
Male	178 (70.9)
**Medical history**	**Frequency (%)**
No	91 (36.3)
Cardiovascular diseases	83 (33.1)
Respiratory diseases	23 (9.2)
Neurologic diseases	31 (12.4)
Diabetes	23 (9.2)
Hypertension	86 (34.3)
**Smoking**	**Frequency (%)**
Yes	59 (23.5)
No	192 (76.5)
**Chief Sleep Complaints**	**Frequency (%)**
Snoring	190 (75.6)
Insomnia	32 (12.3)
Excessive daytime sleepiness	13 (5)
Sleepwalking	5 (1.9)
Nightmare	4 (1.5)
Nocturia	4 (1.5)
Morning headache	3 (1.2)

OSA: obstructive sleep apnea; BMI: body mass index.

**Table 2 life-11-00368-t002:** Descriptive statistical indices and correlation coefficients between demographic, anthropometric, and sleep- and breathing-related dimensions (both subjective and objective measures).

**A**										
**Dimensions**										
		**1 Age**	**2 BMI**	**3 Neck**	**4 Waist**	**5 Daytime Sleepiness**	**6 Sleep Quality**	**7 Risk OSAa**	**8 TST**	**9 Wakenings**
1	Age	-	.16 *	−.06	.11	.12	.11	.13 *	−.27 **	.19 *
2	BMI		-	.33 ***	.58 ***	.15 *	−.16 *	.23 ***	−.17 *	.07
3	Neck circumference			-	.42 ***	.13 *	−.12 *	.04	.00	.00
4	Waist circumference				-	.13	−.07	.23 **	−.05	.05
5	Daytime sleepiness					-	−.04	.63 ***	−.07	.08
6	Sleep quality						-	.04	−.04	.08
7	Risk OSAS							-	.00	.02
8	TST								-	−.69 ***
	Statistics									
	M (SD)	49.50 (12.70)	30.18 (5.07)	40.23 (3.58)	104.17 (13.14)	8.96 (5.00)	7.62 (4.77)	4.69 (1.43)	6.25 (1.16)	5.08 (4.36)
**B**										
**Dimensions**										
		**10 Sleep efficiency**	**11 N1**	**12 N2**	**13 N3**	**14 REM**	**15** **AHI**	**16 RDI**	**17 SpO_2_**	**18 SI**
1	Age	−.27 ***	.00	.04	−.13	.09	.20 ***	.19 **	−.32 ***	−.05
2	BMI	−.23 ***	.02	.01	−.21 **	.16 **	.50 ***	.50 ***	−.38 ***	.28 ***
3	Neck circumference	.05	.29 ***	−.18 **	−.04	−.15 *	.20 **	.23 ***	.07	.12
4	Waist circumference	−.09	.10	.02	−.25 **	.07	.29 ***	.30 ***	−.19 **	.26 ***
5	Daytime sleepiness	−.10	−.01	.02	−.02	.06	.21 ***	.22 ***	−.22 ***	.06
6	Sleep quality	.02	.01	−.06	.00	.13*	−.09	−.08	.02	−.09
7	Risk OSAS	−.04	.02	.05	−.15	.00	.25 ***	.24 ***	−.17 ***	.18 ***
8	TST	.91 ***	.20 ***	−.18 ***	.18 ***	−.28 ***	−.12	−.07	.26 ***	−.17 **
9	Wakenings	−.78 ***	.07	.02	−.23 ***	.15 **	.05	.03	−.17 **	.13 *
10	Sleep efficiency	-	.16 *	−.17 **	.26 **	−.32 **	−.14 **	−.09	.29 **	.12
11	N1 (%)		-	−.76 ***	−.09	−.32 **	−.02	.07	.12	.03
12	N2 (%)			-	−.39 ***	−.02	.05	−.05	−.06	.08
13	N3 (%)				-	−.23 ***	−.19 ***	−.16 ***	−.02	−.09
14	REM (%)					-	.21 **	.19 **	−.10	−.14 *
15	AHI						-	.97 ***	−.32 ***	.28 ***
16	RDI							-	−.33 **	.28 **
17	SpO_2_ (mean)								-	−.05
18	Snoring index									-
	Statistics									
	M (SD)	82.19 (13.21)	39.83 (22.11)	33.03 (22.99)	15.48 (14.82)	11.35 (11.76)	29.62 (20.61)	33.91 (21.56)	89.96 (6.91)	2.17 (0.99)

Note: OSA = obstructive sleep apnea; risk of OSA = STOP-Bang questionnaire; TST = total sleep time; WA = wakenings; SE = sleep efficiency; REM = rapid eye movement sleep; AHI = apnea/hypopnea index; RDI = respiratory disturbance index; SpO_2_ = peripheral oxygen saturation; SI = snoring index. * = *p* < .05; ** = *p* < .01; *** = *p* < .001.

**Table 3 life-11-00368-t003:** Descriptive statistical indices of distribution of gender, tobacco-smoking status, hypertension, cardiovascular diseases, and diabetes between individuals with mild, moderate, and severe AHI (apnea/hypopnea index).

		AHI			
		Mild	Moderate	Severe	X^2^-Test
Gender	Female	23	24	26	X^2^(N = 251, df = 2) = 0.48
	Male	55	52	71	
Smoking	Yes	15	22	22	X^2^(N = 251, df = 2) = 2.08
	No	63	54	75	
Hypertension	Yes	22	18	46	X^2^(N = 251, df = 2) = 12.51 **
	No	56	58	51	
Cardiovascular diseases	Yes	22	20	41	X^2^(N = 251, df = 2) = 6.11 *
	No	56	56	56	
Diabetes	Yes	6	8	9	X^2^(N = 251, df = 2) = 0.34
	No	72	68	88	

Note: AHI = apnea/hypopnea index. * = *p* < .05; ** = *p* < .01.

**Table 4 life-11-00368-t004:** Comparison of anthropometric and polysomnographic characteristics of OSA by severity.

	AHI (Categories)					
	Mild	Moderate	Severe	F-Tests	Partial Eta-Squared	Post Hoc Comparisons
	N 78	75	97			
	Mean (SD)	Mean (SD)	Mean (SD)			
Age (years)	47.10 (13.48)	47.63 (13.14)	52.73 (11.03)	5.54	.043 (S)	severe > mild
BMI	27.12 (4.22)	29.98 (3.79)	32.80 (5.19)	34.29	.217 (L)	severe > moderate > mild
Neck circumference (cm)	39.78 (3.62)	39.68 (3.56)	41.04 (3.45)	4.10	.032 (S)	severe > mild
Waist circumference (cm)	100.09 (11.05)	102.49 (11.43)	108.93 (14.44)	11.68	.086 (L)	severe > mild
PSQI	8.67 (5.16)	6.95 (4.21)	7.23 (4.71)	3.01	0.24 (L)	severe = moderate = mild
ESS	8.21 (4.87)	8.52 (5.15)	9.96 (4.89)	3.15	0.25 (L)	severe = moderate = mild
STOP-Bang	4.41 (1.33)	4.48 (1.58)	5.10 (1.29)	6.61	0.051 (S)	Severe > mild
RDI (event/hour)	14.37 (6.38)	25.77 (6.33)	56.04 (16.95)	301.31	0.70 (L)	severe > moderate > mild
REM stage (%)	9.28 (8.41)	9.75 (10.61)	14.24 (14.17)	4.99	0.39 (L)	severe > mild
Stage N1 (%)	38.20 (21.79)	39.33 (21.26)	41.20 (23.02)	0.41	0.003 (T)	severe = moderate = mild
Stage N2 (%)	35.46 (23.37)	31.71 (23.25)	32.31 (22.58)	0.60	0.005 (T)	severe = moderate = mild
Stage N3 (%)	16.63 (14.78)	18.58 (14.87)	12.12 (14.35)	4.47	0.035 (S)	Moderate>severe
Total sleep time (h)	6.31 (1.15)	6.36 (1.21)	6.09 (1.11)	1.41	0.011 (T)	severe = moderate = mild
Sleep efficiency (%)	83.85 (12.6)	83.52 (13.9)	79.82 (12.9)	2.50	0.20 (L)	severe = moderate = mild
Wake index	4.83 (4.20)	4.80 (4.85)	5.52 (4.09)	0.76	0.006 (T)	severe = moderate = mild
Mean SpO_2_(%)	92.37 (2.99)	90.61 (6.95)	87.55 (8.30)	11.91	0.083 (M)	severe > mild
Snore index	224.37 (167.04)	286.50 (196.12)	341.34 (199.50)	8.29	0.63 (L)	severe > mild

Note: Degrees of freedom for F-tests: (2, 247). OSA: obstructive sleep apnea; BMI: body-mass index; PSQI: Pittsburgh sleep-quality index; ESS: Epworth sleepiness scale; AHI: apnea/hypopnea index; RDI: respiratory-disturbance index.

**Table 5 life-11-00368-t005:** Multiple regression analysis to predict AHI (apnea/hypopnea index), with sociodemographic, anthropometric, and subjective and objective sleep dimensions as predictors.

Dimension	Variables	Coefficient	Standard Error	Coefficient β	t	p	R	R^2^	Durbin–Watson
AHI	Intercept	−.085	2.026	-	−.042	.967	.970	.943	2.085
	RDI	.925	.015	.969	62.38	.000			
	N2 sleep (%)	.078	.014	.087	5.524	.000			
	Total sleep time (h)	−.677	.281	−.038	−2.41	.017			

Note: AHI = apnea/hypopnea index; RDI = respiratory-disturbance index. Excluded variables: age, gender, BMI, neck- and waist circumference, daytime sleepiness, subjective OSA (STOP-Bang), subjective sleep disturbances (PSQI); sleep efficiency, average SpO_2_, snore index, REM sleep, and N1 and N3 sleep (%). All ts < 1.0; ps > .30.

## Data Availability

Data are available upon request to experts in the field and upon thorough scientific justification.

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
