# Peer review of "Compared to Individuals with Mild to Moderate Obstructive Sleep Apnea (OSA), Individuals with Severe OSA Had Higher BMI and Respiratory-Disturbance Scores"

_life, 2021, doi:10.3390/life11050368_

Round 1
Reviewer 1 Report
This is a weel written manuscript. It provides novel, interesting findings on obstructive sleep apnea, an important health problem.
I only ask for correction of some typing errors.
-Abstract, line 8 up - please add p to sleep,
-Introduction, 3rd § , line 3 - probably "and sickest" is sufficient once, not double.
-2.2.4 , line 2. Please replace "is" by hyphen.
Author Response
We thank the Reviewer for the valuable and encouraging comments. Please find the detailed point-by-point-response attached as a separate file. Again, thank you very much for the care devoted to review the manuscript.

Reviewer 2 Report
Manuscript entitled „Compared to individuals with mild to moderate Obstructive Sleep Apnea (OSA), individuals with severe OSA had higher BMI and respiratory disturbances scores” reports on associations between demographic, anthropometric and PSG parameters.
The introduction should be rewritten. Why is cortisol so widely described and is analyzed in the study. Expand on the comorbidities of OSA: cardiovascular (doi: 10.3389/fneur.2018.00635, doi: 10.1183/09031936.00027406), immunological disorders (doi: 10.5664/jcsm.6908, doi: 10.2174/18715281113129990051), respiratory (doi: 10.1097/MCP.0b013e32834317bb, 10.3389/fimmu.2019.00692), diabetes mellitus (doi: 10.3389/fphys.2020.01035, doi: 10.11622/smedj.2017027), hormonal disorders (doi: 10.1016/j.metabol.2018.03.008,10.5603/PiAP.2016.0038). This will provide Additionally, considering the form of the analysis, it should be expanded for more multifaceted analysis, please describe and include in the study possible phenotypes such as positional and REM associated OSA (doi: 10.5664/jcsm.7166, 10.1038/s41598-019-56478-9, 10.1016/j.smrv.2016.10.002). This would greatly enhance the study.
In introduction while stating that 5 breathing related events please name all 5 (only 3 are in the brackets). In the following sentence OAS instead of OSA is written.
Figure 1 has to be adjusted, it is not aligned, the arrow for exclusion should be directed the other way.
Full sentences should be used throughout the manuscript instead of sentence equivalents, which are hard to follow. For example, in methods “Demographic and Anthropometric Data”.
ESS in general consists of 8 items, it is unlikely that standardized tests consist of different number of items in farsi.
Was the distribution of data assessed? Information on data distribution should be stated as the presentation of data depend on its normality. Was correction for multiple comparisons include as with so many parameters it should be used. Were parameters other that significantly correlated taken under consideration in multiple regression?
In results sentence “Associations between age, BMI, subjective and objective sleep parameters and subjective and objective breathing-related parameters” is not finished.
When you split table in two (Table 2, do not point it out, instead name it as 2A and 2B).
Naming all the correlations in the list is unacceptable. Present it in graphical or tabular form.
Considering the form of the analysis, it should be expanded for more multifaceted analysis, adding analysis of phenotypes – positional or REM connected phenotype.
While analyzing objective and subjective data in OSA patients please discuss it in context of healthy individuals (doi: 10.5664/jcsm.7036).
Author Response

(The authors gave the same response as above.)

Round 2
Reviewer 2 Report
Authors adressed all the comments. Please note that in given pdf the figure is still misaligned (the lines and the arrows), therefore if accepted fo the publication it should be corrected.